# Comparative Evaluation of Adipokine Metrics for the Diagnosis of Gestational Diabetes Mellitus

**DOI:** 10.3390/ijms25010175

**Published:** 2023-12-22

**Authors:** Maciej Kamiński, Radzisław Mierzyński, Elżbieta Poniedziałek-Czajkowska, Agata Sadowska, Maciej Sotowski, Bożena Leszczyńska-Gorzelak

**Affiliations:** Chair and Department of Obstetrics and Perinatology, Medical University of Lublin, 20-954 Lublin, Poland; drkaminski1993@gmail.com (M.K.); agata.sadowska@yahoo.com (A.S.); maciek13011@wp.pl (M.S.); bozena.leszczynska-gorzelak@umlub.pl (B.L.-G.)

**Keywords:** gestational diabetes mellitus, adipokines, adiponectin, chemerin, lipocalin, apelin

## Abstract

Gestational diabetes mellitus (GDM) is one of the most common medical disorders in pregnancy. Adipokines, predominantly secreted by adipose tissue, are involved in numerous metabolic processes. The exact role of adipokines in the pathogenesis of GDM is still not well known, and numerous adipokines have been analysed throughout pregnancy and proposed as biomarkers of GDM. This study aimed to evaluate serum adiponectin, chemerin, lipocalin and apelin levels in GDM and non-GDM women, to assess them as clinically useful biomarkers of the occurrence of GDM and to demonstrate the correlation between the levels of the above adipokines in the blood serum and the increased risk of the development of GDM. The role of these adipokines in the pathogenesis of GDM was also analysed. The statistically significant differences between the levels of adiponectin (7234.6 vs. 9837.5 ng/mL, *p* < 0.0001), chemerin (264.0 vs. 206.7 ng/mL, *p* < 0.0001) and lipocalin (39.5 vs. 19.4 ng/mL, *p* < 0.0001) were observed between pregnant women with GDM and healthy ones. The diagnostic usefulness of the tested adipokines in detecting GDM was also assessed. The research results confirm the hypothesis on the significance of adiponectin, chemerin, lipocalin and apelin in the pathophysiological mechanisms of GDM. We speculate that these adipokines could potentially be established as novel biomarkers for the prediction and early diagnosis of GDM.

## 1. Introduction

Gestational diabetes (GDM) is one of the most common medical disorders occurring during pregnancy. According to global data, GDM affects about 14% of pregnant women worldwide [1,2]. GDM is defined as glucose intolerance that is first developed or diagnosed during pregnancy [3]. It is correlated with an increased risk of complications during pregnancy for both the mother and the foetus. Possible complications include macrosomia, shoulder dystocia, foetal hypoglycemia, and increased caesarean section rate. In addition, women with GDM during pregnancy are at risk of developing cardiovascular diseases and type 2 diabetes mellitus (T2DM) in later life [4].

It is believed that GDM development results from β-cell dysfunction against a background of chronic insulin resistance observed in pregnant patients [1]. The β-cells dysfunction is caused by prolonged, excessive insulin production in response to chronic excess glucose [5].

Abnormalities in β-cells can occur at various stages: proinsulin synthesis, post-translational modifications, blood glucose sensing, and the exocytosis process. Most of the susceptibility genes that are correlated with GDM are associated with β cells, including KQT-like voltage-gated potassium channels 1 (Kcnq1), and glucokinase (GCK) [6].

In the early stages of pregnancy, inulin sensitivity does not change from pre-pregnancy levels. As pregnancy progresses, insulin sensitivity decreases; initially, it is more pronounced in muscle tissue than adipose tissue. Decreased maternal insulin sensitivity in women with GDM could be associated with a dysregulated insulin signalling pathway [7,8]. During pregnancy, the increase in demand for insulin due to insulin resistance is compensated by its increased production, even by 50%. In the case of GDM, however, the increase in insulin secretion is much lower, which results in the development of hyperglycaemia [9,10]. In the postpartum period, within a few days after delivery of the placenta, there is a rapid and significant increase—approximately 120%—in insulin sensitivity compared to late pregnancy [11].

The importance of adipokines in the pathophysiology of GDM is still not fully understood. The dysregulation of the metabolism and/or placental function of several adipokines may play a key role in the pathogenesis of GDM [12,13]. Adipokines are involved in numerous metabolic processes, such as insulin secretion, insulin sensitivity, distribution of adipose tissue, regulation of appetite, development of inflammation and control of the body’s energy management. The dysregulation of several adipokines metabolism, such as adiponectin, leptin, AFABP (adipocyte-specific fatty acid binding protein), tumour necrosis factor α (TNF-α), nesfatin, vaspin, chemerin and lipocalin, seems to be a potential factor involved in the pathophysiology of GDM [13,14,15]. Numerous adipokines have been studied as markers for GDM. However, no biomarker has been confirmed for GDM diagnosis [16].

Adiponectin, a close homolog of the complement family 1q (C1q), is a 30 kDa monomeric glycoprotein. It is produced by the *AdipoQ* gene and shares structural similarities with TNFα, collagen VIII and IV, and the C1q complement protein. Its structure is composed of an N-terminal signal sequence, a non-homologous or hypervariable region domain collagen and C-terminal C1q-like globular domain [17]. Peroxisome proliferator-activated receptor (PPAR)-γ upregulates the expression of *AdipoQ* [18], while pro-inflammatory signals such as TNF-α downregulate its expression, and *AdipoQ* hypermethylation has been correlated with modified gene expression and glucose metabolism [19]. Adiponectin secretion is modified by the molecular chaperones ERp44, Ero1-Lα and DsbA-L in the endoplasmic reticulum. Two separate types of adiponectin receptors, AdipoR1 and AdipoR2, have been described [20].

Adiponectin enhances the action of pancreatic β cells to lower systemic glucose levels by inhibiting hepatic glycogenolysis [21]. Its levels are inversely associated with fasting glucose and insulin levels, body fat, and insulin resistance in adults. The mean plasma adiponectin concentration is reduced in patients with T2DM and obesity. The circulating adiponectin levels decrease correspondingly throughout pregnancy in correlation with increasing insulin resistance and insulin sensitivity, and adiponectin levels are low in patients with GDM [22]. The low concentrations of adiponectin in all trimesters of pregnancy are correlated with an increased risk of metabolic disorders observed in pregnant patients, higher frequency of GDM and higher risk of adverse perinatal outcomes [23]. The decreased concentration of adiponectin in the first trimester of pregnancy or prior to pregnancy is supposed to predict GDM development [23].

Lipocalin-2 (LCN2), also known as neutrophil gelatinase-associated lipocalin (NGAL) is a 178 amino acid protein that has 3 molecular forms: a 25 kDa monomer, a 45 kDa homodimer and a 135 kDa heterodimer in complex with the matrix metalloproteinase 9 (MMP-9) [24]. Because of ligand-binding, post-translational modifications and protein–protein interactions, various variants performed various functions [25]. The solute carrier family 22 member 17 and the megalin/glycoprotein GP330 were described as two different receptors for lipocalin-2 [26]. It has been suggested that LCN2 is a potential mediator linking chronic low-grade inflammation with obesity. It has also been proven that the level of LCN2 is closely related to insulin resistance and hyperglycaemia [27]. Inflammatory features occurring in diabetes and obesity correlated with elevated LCN2 levels in blood plasma and some tissues in both humans and laboratory animals suggest a pro-diabetogenic and pro-obesity function of LCN2 [28].

Contrarily, however, numerous studies indicated the anti-diabetogenic and anti-obesity properties of LCN2 [29]. Amid these conflicting findings, no experimental evidence exists to clarify why LCN2 exhibits such contrasting roles in different studies [30]. Kobara and co-authors have suggested that the placenta and trophoblast are the primary sources responsible for elevated plasma LCN2 levels in pregnant women [31]. These data implicate that LCN2 plays a significant role in glucose homeostasis, and may be considered as a possible marker of glucose metabolism disorders in pregnancy.

Chemerin is originally synthesized as the 163 amino acid chemerin preproprotein. Prochemerin becomes bioactive through serial processing by single proteases (e.g., cathepsin G, elastase, plasmin, tryptase) or multiple serine proteases [32]. Chemerin also binds to two other receptors, chemerin receptor 2 (chem2, also known as GPR1) and chemokine receptor-like 2 (CCRL2) and is almost exclusively expressed and synthesized in white adipose tissue [33]. Both of them are members of subclass A8 belonging in the rhodopsin group of G protein-coupled receptors (GPCRs). CCRL2 does not encode a full serpentine G protein-coupled receptor GPCR and does not appear to be a signalling receptor but binds chemerin and may play a role in presenting it to the signalling receptors chem1 and chem2 [34]. It has been described that chemerin concentrations increase throughout decidualization and can affect NK cell accumulation and vascular remodelling in the first trimester of pregnancy, which seems crucial during placentation [35]. Chemerin affects carbohydrate metabolism. As a result of the action of chemerin on muscle tissue cells, glucose uptake is inhibited, and insulin sensitivity is reduced [36]. Serum chemerin levels are strongly correlated with symptoms of the metabolic syndrome, such as obesity and insulin resistance. It has been suggested that chemerin may play a key role in the pathogenesis of GDM [33,37].

Adiponectin and chemerin are referred to as contrary adipokines [38]. Adiponectin is one of the adipocyte-specific proteins with novel metabolic applications by improving insulin sensitivity and modulating glucose and fatty acid catabolism, while chemerin is thought to inhibit insulin signalling and glucose catabolism. Both of these adipokines are also involved in the coordination of reproductive activities but have opposing functions [39]. It has been published numerous studies analysing the role of adiponectin and chemerin in insulin resistance and metabolic syndrome. It has been also noticed that plasma adiponectin levels correlated negatively, while chemerin correlated positively with obesity and insulin resistance. Also, some authors suggest that a high chemerin/adiponectin ratio plays an important role in causing dyslipidaemia and metabolic syndrome in patients with impaired metabolism [40].

The dysregulated adiponectin to chemerin ratio during various metabolic disorders makes it truly worthy of therapeutic use. The key role of adiponectin in modulating glucose metabolism and insulin sensitivity emphasize its potential as a promising therapeutic target. Gene therapy may probably increase adiponectin expression or activity in diabetic patients [40]. However, this requires further research.

Apelin is produced as a pre-pro-peptide of 77 amino acids and shorter active forms with 12, 13, 16, 17 and 36 amino acids. Apelin-13 is probably the most biologically active form. In humans, the apelin coding sequence is located on chromosome Xq25-q26.1. [41,42]. Apelin acts on the peripheral and central nervous systems, which are involved in enzymatic response, hemodynamic homeostasis, angiogenesis, vasodilation, glucose metabolism and the development of atherosclerosis associated with oxidative stress [41,43]. Apelin has been reported to be present in placental tissue, suggesting a role for this peptide during pregnancy [44]. The high expression of apelin in the placenta can suggest that its fetoplacental activity is modulated in a paracrine or autocrine way. However, the specific signals regulating its accessibility are poorly understood [45]. During pregnancy and lactation, apelin expression in the mammary glands can increase 7 to 20 times [46,47]. In humans, the evidence supporting the role of apelin in glucose metabolism remains controversial. Some studies find elevated apelin levels in very small populations of obese individuals with impaired glucose tolerance or T2DM. At the same time, other authors have reported surprisingly low levels of apelin in a group of obese individuals with newly diagnosed T2DM compared to healthy individuals without diabetes [48]. For this reason, further studies need to be performed to examine the role of apelin in metabolic disorders.

There are more than 50 different adipokines known to function in various aspects of diabetes and associated complications. However, not all appear to play an important role in the pathogenesis of GDM. After a thorough literature review and in view of the recent evidence, we focus on the one of the most well-known adipokine: adiponectin, and the not so well-studied: LCN2, chemerin and apelin. The exact role of adiponectin, LCN2, chemerin and apelin in the pathophysiology of GDM still remains not fully clear and the relationship between circulating concentrations of these adipokines and risk of GDM is not fully known.

The aim of this study is to determine serum adiponectin, chemerin, lipocalin 2 and apelin levels in GDM patients, assess the correlation between these adipokines and discuss their possible role in the diagnosis and pathogenesis of gestational diabetes mellitus.

## 2. Results

### 2.1. Characteristics and Comparison of the Study Group (GDM) and the Control Group

The study group consisted of 90 pregnant patients diagnosed with GDM. The control group consisted of 84 healthy pregnant women without diabetes. There was a statistically significant difference in median BMI at the first prenatal visit and at 24–28 weeks of pregnancy between women in the study and control groups (respectively: BMI at the first prenatal visit 23.0 vs. 22.5, *p* = 0.0356, Figure 1A; BMI at 24–28 weeks of pregnancy 26.5 vs. 26.0, *p* = 0.0337, Figure 1).

A statistically significant difference was observed in glycaemia determined before the consumption of 75 g of glucose on an empty stomach (TTG0) and measured after 1 h (TTG1) and after 2 h (TTG2) between women in the study and control groups (TTG0: 94.0 vs. 81.0 mg/dL, *p* < 0.0001; TTG1: 185.0 vs. 138.0 mg/dL, *p* < 0.0001, TTG2: 155.0 vs. 127.5 mg/dL, *p* < 0.0001). Detailed data showing the comparison of clinical characteristics of the study and control groups are presented in Table 1.

The concentrations of the tested adipokines were compared between the study group and the control group. Significant differences in the concentrations of adiponectin (respectively: 7234.6 vs. 9837.5 ng/mL, *p* < 0.0001), chemerin (respectively: 264.0 vs. 206.7 ng/mL, *p* < 0.0001) and lipocalin (respectively: 39.5 vs. 19.4 ng/mL, *p* < 0.0001) were found between the groups. A trend towards significance was observed when comparing the apelin concentration between the test and control group (11,392.0 vs. 9364.0 pg/mL, *p* = 0.0554). Detailed data showing the comparison of the concentrations of the tested adipokines are presented in Table 2. The correlations between adiponectin, chemerin, LCN2, apelin and selected variables are presented in Table 3, Table 4, Table 5 and Table 6.

### 2.2. Evaluation of the Diagnostic Usefulness of the Tested Adipokines in Detecting GDM Diabetes Using ROC Curves

The diagnostic usefulness of the tested adipokines in detecting GDM was assessed. Adiponectin (cut-off: ≤8482.06 ng/mL) had a sensitivity of 81.11% and a specificity of 82.14% in detecting GDM (AUC 0.801, 95%CI: 0.733–0.857; *p* < 0.0001, Figure 2). Chemerin (cut-off: >249.99 ng/mL) had a sensitivity of 66.67% and specificity of 89.29% in detecting GDM (AUC 0.767, 95%CI: 0.697–0.827; *p* < 0.0001, Figure 2). Lipocalin (cut-off: >23.69 ng/mL) had a sensitivity of 86.67% and specificity of 71.62% in detecting GDM (AUC 0.887, 95%CI: 0.830–0.930; *p* < 0.0001, Figure 2). Apelin (cut-off: >9888.2 pg/mL) had a sensitivity of 61.11% and specificity of 58.33% in detecting GDM (not statistically significant, AUC 0.584, 95%CI: 0.507–0.658; *p* = 0.0529, Figure 2).

Detailed data on the diagnostic usefulness of the tested adipokines in detecting GDM using ROC curves are presented in Table 7.

One-way logistic regression analysis revealed a significant increase in the risk of gestational diabetes mellitus (GDM) with a rise in chemerin concentration by 10 ng, LCN2 by 1 ng, and apelin by 1 ng (1000 pg). The observed increases were 20% (OR = 1.20, 95%CI: 1.12–1.29; *p* = 0.0001), 18% (OR = 1.18, 95%CI: 1.12–1.24; *p* < 0.0001), and 6% (OR = 1.06, 95%CI: 1.01–1.11; *p* = 0.0257), respectively. The same analysis, in turn, showed that an increase in adiponectin concentration by 1 μg/mL (1000 ng/mL) was associated with a significantly lower (by 26%) risk of GDM (OR = 0.74, 95%CI: 0.67–0.83).

## 3. Discussion

The prevalence of GDM has been increasing markedly in recent years. This is mainly due to the increasing number of obese women and the increase in the average age of mothers [49,50]. Although GDM usually resolves shortly after delivery, GDM patients have a substantially higher risk of developing many complications in their future life. In addition, the risk of developing both obesity and T2DM and cardiovascular disease in their offspring increases. GDM, therefore, contributes to the development of an intergenerational vicious circle of obesity and diabetes, which affects the population’s health as a whole [49,51,52].

There is an urgent need to find ways to early detect and prevent the development of GDM. Appropriate management of pregnant women with GDM still raises many unanswered questions and preventing the progression of GDM to type 2 diabetes in the postpartum period remains a challenge [11].

Due to the problems associated with GDM, the HAPO (Hyperglycaemia and Adverse Pregnancy Outcome), a study was conducted to clarify the impact of maternal hyperglycaemia on the occurrence of pregnancy complications. This study analysed the relationship between the fasting insulin level and the 1st and 2nd hour of the OGTT and the risk of complications. Primary complications included caesarean section, macrosomia, hypoglycaemia or hyperinsulinism in neonates. The group of secondary complications included polycythaemia, hyperbilirubinemia, respiratory distress syndrome and shoulder dystocia. As maternal glucose levels increased, the incidence of each of the above complications was observed. However, neonatal hypoglycaemia was the least affected. A powerful relationship was shown between the glucose level in the mother’s blood and the child’s weight [53,54,55].

Numerous adipokines have been demonstrated in GDM. However, the role of adipokines in the pathophysiology of GDM is still not well explained. Excessive activation of the adipose tissue could play a crucial role in the mechanisms leading to GDM development [56]. Monocytes and macrophages derived from adipose tissue are the source of inflammatory factors [57,58].

In the conducted study, we showed a statistically significant difference in the level of adiponectin in the blood serum of pregnant women. In the GDM group, the level of adiponectin was lower compared to the control group. We also found that adiponectin levels correlated with BMI at the first prenatal visit and BMI at sampling in the GDM and control groups. In addition, we demonstrated the diagnostic usefulness of the adiponectin level test in detecting gestational diabetes. Adiponectin was 81.11% sensitive and 82.14% specific in detecting GDM.

Similar results were obtained in a study by Bozkurt et al. Lower levels of adiponectin have also been shown in GDM patients compared to pregnant women without GDM. Moreover, for lower gestational age, these differences were more significant. This study confirmed the usefulness of adiponectin determination in predicting the development of GDM, especially before the 21st week of pregnancy [21].

The study conducted by Atarod et al. in patients between 24 and 28 weeks of pregnancy also showed a significantly lower level of adiponectin in the group of pregnant women with GDM compared to patients without GDM [59]. The study by Mierzyński et al. also performed between 24 and 28 weeks of gestation showed a significant difference between the concentration of adiponectin and pre-pregnancy BMI. Compared to the non-diabetic patients, median adiponectin levels were significantly lower in patients with GDM (*p* < 0.01) [60].

Saini et al. compared adiponectin levels between pregnant women with and without GDM. This study also produced similar results to our study. The level of adiponectin in the serum of women with GDM was significantly lower (5.43 ± 2.28 µg/mL) relative to the control group (13.03 ± 5.53 µg/mL) [61]. In a meta-analysis of 15 studies and 17 comparisons conducted by Xu et al., the results of serum adiponectin measurements in 560 GDM patients and 781 controls were significantly different. In patients with GDM, the level of adiponectin was lower (*p* < 0.00001) [62].

Lacroix et al. examined pregnant women in the first trimester of pregnancy. They showed that women who developed GDM had lower adiponectin levels in the first and second trimesters than women without GDM [63]. The same result was also obtained in a study performed by Sweeting et al. on a group of 980 patients [64]. In addition, Lacroix et al. revealed a relationship between lower adiponectin levels in the first trimester and the prevalence of obesity. Pregnant women with lower levels of adiponectin in the first trimester have been shown to have increased levels of insulin resistance and a greater likelihood of developing GDM regardless of obesity [63]. In addition, a study by Lacroix showed using logistic regression that a reduction in adiponectin levels of 1 µg/mL in the first trimester was associated with an increased risk of developing GDM (OR 1.14 [95% CI 1.04–1.25] [46]. Williams et al. presented a study in which they showed that a decrease in the adiponectin level by every µg per ml increases the risk of GDM by 20% [47]. Our study showed that an increase in adiponectin concentration by 1 µg (1000 ng/mL) was associated with a significantly lower (by 26%) risk of GDM.

It has been noticed that adipose tissue from women with gestational diabetes was found to have decreased concentrations of adiponectin, even when considering BMI [19]. It has been also described that hypermethylation of the human adiponectin gene (*ADIPOQ*) was correlated with obesity [65]. Ott et al. described hypermethylated regions of *ADIPOQ* in white adipose tissue from patients with GDM [19]. They also found significant changes in adipose tissue *ADIPOQ* methylation in children intrauterinely exposed to GDM. This may indicate both tissue-specific and heritable epigenetic regulation of adiponectin expression by exposition to GDM, even when considering maternal BMI and the gender of the offspring [19]. Adiponectin is considered an adipokine that increases insulin sensitivity with an additional anti-inflammatory effect. Mechanisms that may explain the effect of increasing insulin sensitivity include increased sensitivity of insulin receptors, decreased gluconeogenesis, increased lipid oxidation and inhibition of the tumour necrosis factor-alpha (TNF-α) signal in adipose tissue [14,66,67]. It has also been noticed that adiponectin influences ‘normal’ adipocyte expansion and promotes an anti-inflammatory phenotype in adipose tissue. In animal models and clinical studies, the adiponectin concentrations are decreased in GDM, while inflammatory parameters such as TNF-α and interleukin (IL)-6 simultaneously increase [68]. Considering the role of adiponectin in maintaining normal function of adipose tissue, decreased adiponectin levels during pregnancy may lead to the development of inflammatory and insulin-resistant adipose tissue phenotypes in GDM.

Adiponectin receptors have been found in β-cells, implying functional adiponectin signalling [69]. Adiponectin signalling could also play a role in the maintenance of β-cell mass and, probably, activity [70]. During gestation, adiponectin was also noticed to influence β-cell expansion by influencing the expression of the lactogen in the placenta [71]. These findings suggest a potential role of adiponectin in structural β-cell adaptations to pregnancy and the pathogenesis of GDM.

Our study showed statistically significantly increased serum LCN2 levels in GDM patients compared to non-GDM patients. Regarding diagnostic utility, LCN2 was 86.67% sensitive and 71.62% specific in detecting GDM.

Increased levels of LCN2 were noticed in metabolic disturbances such as obesity, T2DM, hypertriglyceridemia, hyperglycaemia, PCOS and preeclampsia [30,72]. LCN2 expression begins in the embryonic period. Upregulation of LCN2 is modulated by phosphatidylinositol 3-kinase and mitogen-activated protein kinase. An increased concentration of LCN2 is correlated with higher BMI, fasting plasma glucose, fasting plasma insulin, HOMA-IR, triglycerides, total cholesterol, hs-CRP in GDM patients, and there is an inverse correlation with HDL-cholesterol or LDL-cholesterol [27].

Sweeting et al. conducted the study in the first trimester of pregnancy in a large group of 980 women and compared the level of LCN2 between the control group and patients with GDM. In their study, similarly to our findings, higher levels of LCN2 in the group of women with GDM were observed [64]. There was a 10% higher LCN2 level in the GDM group, and the researchers suggested that minor differences probably reflect influence of ethnicity on biomarker correlations with GDM. In their first trimester GDM risk prediction model, the combination of triglycerides, pregnancy-associated protein A and LCN2 performed best in the examined cohort and was utilized in the final model. The performance (AUC) of the model in predicting GDM detection overall was 0.91 (95% CI 0.89–0.94), and for early GDM it was 0.93 (95% CI 0.89–0.96). They also noticed that some studies have revealed the correlation between reduced adiponectin concentrations and GDM [73], but in their study, LCN2 was the best-performing adipokine for GDM prediction. However, this can be a result of incomplete data on LCN2 [62].

The study performed by Yin et al. in the third trimester of pregnancy revealed that the levels of LCN2 and TNF-α in the serum of pregnant women were significantly increased in women with GDM than the healthy ones. In contrast, LCN2 levels were higher in women with GDM in cord blood testing, while TNF-α levels were higher in mothers without GDM. In addition, expression levels of LCN2 and TNF-α were shown to be significantly higher in women with GDM than in control group, both in placental and umbilical cord samples. In addition, serum LCN2 levels in mothers with GDM were positively correlated with LCN2 levels in placentas. These data suggest that higher placental expression of LCN2 can be responsible for its high serum concentrations in GDM patients, thus influence the degree of insulin resistance [74].

Our study revealed a significant but weak positive correlation between the LCN2 concentration and BMI at the first prenatal visit, BMI during pregnancy and TTG0.

Significantly higher LCN2 concentrations in GDM patients, especially in Caucasians with BMI > 25 kg/m^2^, have been found. Lou et al. compared a group of healthy pregnant women with women with GDM in the third trimester of pregnancy, which were additionally divided into two subgroups (BMI below and above 25 kg/m^2^). The serum level of LCN2 in GDM patients, regardless of BMI, was statistically significantly higher compared to patients without GDM. In the case of pregnant GDM with BMI > 25 kg/m^2^, the level of LCN2 was noticeably higher than in the group of pregnant GDM with BMI < 25 kg/m^2^ [75].

The authors of this study also observed the positive correlations between LCN2 and features of insulin resistance: fasting plasma glucose (FGP), HOMA-IR, fasting plasma insulin (FPI), high-sensitivity C-reactive protein (hs-CRP), total cholesterol and triglyceride. Moreover, LCN2 mRNA and protein expression in white adipose tissue were increased in overweight patients. They postulated that LCN2 may influence the progression of insulin resistance in gestational diabetes and its expression in adipose tissue can be correlated with obesity in women with GDM [75]. Mierzynski et al. demonstrated in the second trimester of pregnancy significantly higher levels of LCN2 in GDM patients compared to women without GDM. In addition, similarly to this study, a strong positive correlation between the levels of chemerin and LCN2 was demonstrated [43].

In the presented study, a significant, moderate positive correlation between chemerin and lipocalin concentrations and a negative correlation between adiponectin and lipocalin concentrations was reported. The possible explanation of such relationship and clinical importance is not clear. It could be speculated that these findings might confirm the significance of these adipokines in the pathophysiology of GDM.

Karakaya et al. compared the level of LCN2 in the urine of women with and without GDM in the second trimester of pregnancy. The LCN2 concentration was significantly increased (*p* < 0.014) in GDM women. Higher levels of LCN2 can cause damage to the renal tubules. In addition, they showed a statistically significant positive correlation between LCN2 levels and the level of HbA1c [76]. LCN2 has a complex biological function and is involved in the pathophysiology of several diseases. It is suggested that LCN2 could play a role in pathogenesis of gestational diabetes through influence on inflammation and endothelial cell dysfunction. LCN2 induces several proinflammatory mediators, such as interleukin-6, matrix metalloproteinase 2 (MMP2) and MMP9 [25]. LCN2 modulates the expression of TNF-α in fat tissue, an insulin resistance-inducing factor. It has also been described that LCN2 is critically involved in diet-induced endothelial dysfunction by influencing cytochrome P450 2 C9 activity [77].

However, the exact role of LCN2 in the pathogenesis of GDM and the mechanisms of its action are still not well known, and further studies are needed to determine the role of lipocalin-2 in pathological pregnancies complicated by GDM.

Our study showed a significant difference in chemerin concentration between the study and control groups. Chemerin was 66.67% sensitive and 89.29% specific in detecting GDM.

Chemerin belongs to a group of novel adipokines. It is believed that chemerin has a significant effect on glucose and lipid metabolism regulation. It is the product of the retinoic acid receptor 2 (RARRES2) gene located on chromosome 7 [78]. Its increased level seems to have an influence on some metabolic and cardiovascular disorders and contributes to the development of inflammation [79]. Chemerin concentrations are associated with BMI, overweight, serum lipid levels and blood pressure [80,81]. The chemerin secretion during pregnancy increases significantly with the length of its duration [82]. Some researchers have noticed that maternal peripheral circulating chemerin concentration during pregnancy is determined by maternal obesity but is not affected by GDM [83].

In 2022, Wang et al. published a study on a group of 703 women in first and second trimester of pregnancy and found that plasma chemerin levels are higher in women with GDM (135.8 ± 53.5 ng/mL) compared to pregnant women with normal glucose tolerance (93.4 ± 38.9 ng/mL) and non-pregnant healthy subjects (103.4 ± 45.3 ng/mL). In addition, a positive correlation was found between the concentrations of chemerin and the HOMA-IR index (r = 0.4322, *p* < 0.0001), which suggests an essential role of chemerin in the development of insulin resistance [84]. A meta-analysis by Bellos et al. assessed 11 studies on chemerin and its levels during non-complicated and GDM-complicated pregnancy. Five studies showed a significant increase in chemerin concentrations in GDM women. Only one study revealed decreased chemerin concentrations, and in five studies, no significant correlation was observed [85].

Fatima et al. compared patients with GDM in the late second trimester of pregnancy to healthy ones. The levels of chemerin in serum were significantly higher in the GDM group [93.39 ± 45.43 vs. 14.35 ± 5.88] (*p* < 0.01). The same study showed that for the value of 415.49 ng/mL, chemerin is characterized by a sensitivity of 96% and a specificity of 72% in detecting GDM [86]. In 2019, Wang et al. reported in the second and third trimester of pregnancy the predictive value of chemerin as a GDM biomarker at 73.33% of sensitivity and 76% of specificity [87]. These results correlate with the results of our study. Mierzynski et al. conducted a study comparing pregnant women with and without GDM. Also, this study observed significantly higher levels of chemerin in the GDM group. The above report showed a significant positive correlation between chemerin and LCN2 concentrations both in the GDM group and in the control group [43].

The relationship between GDM and obesity and the chemerin levels was also analysed. It has been found that the level of chemerin in the group of obese patients without GDM (195.0 ± 34.4 ng/mL) or overweight without GDM (151.0 ± 15.5 ng/mL) was significantly higher compared to healthy pregnant women of normal weight (73.1 ± 8.6 ng/mL) [88]. It might suggest that it could correlate with insulin resistance and proinflammatory conditions due to increased levels of inflammatory mediators such as TNF-α, resistin or IL-6 [88].

Our study showed a significant difference in median BMI at the first prenatal visit and during pregnancy between women in the study and the control group. However, in both groups, there were no obese patients before pregnancy and with BMI > 30 kg/m^2^ at sampling. The women with pre-pregnancy diabetes mellitus, insulin resistance recognized before gestation, metabolic disorders (such as PCOS) and any form of hypertension were excluded from our study. So, the risk of increased insulin resistance and higher levels of mediators of inflammation in both groups of our study was relatively small.

It has been also noticed that the level of chemerin in umbilical cord blood, peripheral blood, adipose tissue and placental tissue was statistically significantly higher in the group with GDM than in the control group. In addition, a correlation was shown between blood levels of chemerin and maternal insulin resistance index [81]. The study by Pfau et al. opposes the majority of studies looking for a relationship between the concentrations of chemerin in women with GDM. This study found no significant difference between the chemerin levels in the control and study group. However, a significant relationship was observed between the levels of chemerin, the HOMA index and creatinine levels [89].

The mechanism by which chemerin is involved in endocrine and metabolic regulation is still unclear. Despite the growing evidence supporting an association between chemerin and GDM, the exact mechanisms involved are also not well known. Increased insulin resistance and promotion of subclinical inflammation have been postulated as the most critical pathophysiological mechanisms. Proving the precise mechanisms and link between chemerin and GDM occurrence requires further research. The number of studies is relatively small, and there is a lack of larger-scale research considering different ethnic populations.

Our study revealed slightly higher apelin concentrations in the GDM group, but the difference was insignificant. We also noticed that apelin, compared to the other analysed adipokines, has the lowest sensitivity and specificity in detecting GDM.

Apelin is an endogenous ligand for a specific APJ receptor belonging to the family of G protein-coupled receptors [90]. The physiological role of apelin is still not well known. Apelin plays essential roles in the physiology and pathophysiology of several organs, including regulation of blood pressure, cardiac contractility, angiogenesis, metabolic balance and cell proliferation, apoptosis or inflammation. Apelin is also involved in glucose homeostasis, and higher levels of apelin were observed in obesity and T2DM [91]. In humans and animals, apelin is a stimulant of the glucose transporters (GLUT1/2/4) expression in the muscle and adipose tissue via the PI3K/Akt and AMPK pathways [92]. It has been noticed that the placenta produces significant amounts of apelin, and apelin levels in the foetal blood are higher than in mothers [93]. In animal studies, apelin significantly improves preeclampsia symptoms, impairs endothelial nitric oxide synthase/nitric oxide signalling, and has a positive impact on oxidative stress activation [94].

Higher plasma concentrations of apelin correlated with hyperinsulinemia were observed in animal models of obesity. It has been suggested that apelin excessive production by adipose tissue may be involved in some obesity-related disorders [95]. Proinflammatory adipocytokines can influence the secretion of apelin. It has been found that the apelin levels are increased in insulin resistance. A positive association between apelin and expression of TNF-α in adipose tissue and a direct upregulation of apelin expression in both human and rodent adipocytes by TNF-a were also found [96]. Increased apelin has been suggested to be a compensatory mechanism that inhibits pancreatic secretion, insulin sensitivity and glucose uptake by non-insulin-dependent muscle tissue [97]. Higher plasma apelin levels in T2DM and T1DM prove this compensatory mechanism, which decreases insulin resistance and leads to a decrease in apelin levels. It is suggested that apelin also has anti-diabetic features and may be used as a therapeutic agent for type I and II diabetes [98]. However, this hypothesis requires further research.

Guo et al. compared serum apelin levels in GDM patients and non-GDM patients in the second and third trimesters of pregnancy. In the second trimester, apelin levels were significantly higher in patients with GDM than those without GDM. In the third trimester, the difference in apelin concentrations between the two groups was not statistically significant. The authors of this report concluded that apelin concentrations in GDM patients are higher than in non-complicated pregnancies, increase during pregnancy and reach their maximum in the third trimester [99].

Aslan et al. compared apelin levels in the serum of women in the third trimester of pregnancy, with and without GDM. In GDM women, apelin concentrations were significantly higher (13.5 ± 8.3 vs. 9.6 ± 5.9 ng/mL, *p* = 0.001). However, when comparing the apelin concentrations in umbilical cord blood in both groups, the difference was insignificant (8.8 ± 4.3 vs. 8.2 ± 1.9 ng/mL, *p* = 0.618). Moreover, the concentrations of apelin in maternal blood and umbilical cord blood was negatively correlated with gestational age and birth weight. The reasons for increased apelin levels in GDM are not entirely clear; it may be caused by its increased secretion or decreased metabolism [100]. The opposite results were obtained by Boyadzhieva et al. The study compared the levels of apelin in patients with and without GDM during third trimester of pregnancy and postpartum. During pregnancy, apelin levels were significantly lower in the group of patients with GDM. However, in the postpartum period, these differences were not significant [101].

Telejko et al. performed a study and compared apelin levels in the serum of GDM and non-GDM patients between 24 and 32 weeks of pregnancy and of patients at term. No significant differences were noticed between the two groups. There were also no significant differences between the groups in the levels of apelin mRNA expression in subcutaneous and visceral adipose tissue [45]. The authors also noticed high placental apelin expression and suggested that its fetoplacental activity is modulated in a paracrine or autocrine way. However, the specific signals that regulate its availability are still not explained [45]. Sun et al. performed a meta-analysis of 20 studies, including 1493 GDM patients and 1488 healthy pregnant women. This analysis found no significant differences in circulating apelin levels between the two groups [102]. Kourtis et al. compared pregnant patients between 24 and 28 weeks of pregnancy with non-pregnant patients. In this study, the apelin levels were significantly lower in pregnant women. The authors concluded that apelin may be a marker of oxidative stress in pregnancy [103].

The differences between our results and results published by other researchers can be explained through differences in the study protocols and different groups of examined patients, including the type of GDM with a dietary or insulin treatment and the severity of the disease. On the other hand, feeding conditions, weeks of gestation and place of measurement can influence these differences.

Thus, publications regarding apelin and its relationship to the pathophysiology of GDM are conflicted. At the moment, it is not entirely sure whether apelin might become a useful marker in the diagnosis of GDM in the future. Nevertheless, this issue requires further research.

## 4. Materials and Methods

The study was carried out at the Department and Clinic of Obstetrics and Perinatology of the Medical University in Lublin, Poland. The study involved 90 patients with GDM and 84 healthy pregnant patients. All patients signed an informed consent form before participating in the study. The study was conducted following the principles expressed in the Declaration of Helsinki. The Bioethics Committee of the Medical University of Lublin KE-0254/117/2018 approved the study protocol. The tests were performed as part of routine diagnostic procedures carried out at the Clinic and did not burden the patients in any way.

The inclusion criteria for the study were as follows: singleton pregnancy, gestational age from 24 to 29 weeks, the first medical visit in pregnancy before 10th week and GDM first diagnosed in the current pregnancy before the 28th week of pregnancy.

Patients with at least one of the following criteria were excluded from the study: multiple pregnancy, gestational age before 24 weeks and over 29 weeks, hypertension, pregestational diabetes mellitus, foetal growth restriction, insulin resistance recognised prior to pregnancy, polycystic ovary syndrome, chronic kidney diseases, chronic liver diseases, systemic lupus erythematosus, antiphospholipid syndrome, inflammatory and infectious diseases.

The oral glucose tolerance test (OGTT) with 75 g of glucose was performed in all patients between 24 and 28 weeks of gestation. The patients were diagnosed with GDM after fulfilling at least one from the following criteria: fasting glucose level 5.1–6.9 mmol/l (92–125 mg/dL), glucose at 60 min: ≥10.0 mmol/L (≥180 mg/dL), glucose at 120 min: 8.5–11 mmol/L (153–199 mg/dL).

Information about pregnancy and delivery, maternal and family history, and maternal age was obtained from medical records. Body mass index (BMI) was calculated at the first prenatal visit and at sampling as the ratio of body weight (kg) to height (m^2^). The weight was measured using a digital scale with a maximum capacity of 150 kg. Height was measured with a standard wall tape in a standing position.

All patients included in the study (174 patients) had fasting blood samples taken to determine the concentration of four adipokines: adiponectin, chemerin, lipocalin-2 and apelin. The blood specimens for research analysis were taken at the same time as the blood specimens for routine laboratory analysis. After centrifugation (2000 g/min for 10 min) and dividing the serum into portions, the blood was delivered to the laboratory within 20 min. Serum samples were stored at minus 70 °C until adipokine levels were determined. Determination of adipokine levels was performed using ELISA kits: Human Lipocalin-2/NGAL, BioVendor R&D Products, Czech Republic, Human Chemerin, BioVendor R&D Products, Czech Republic, Human Apelin, Cloud-Clone Corp., Houston, Tex, USA, Human Adiponectin, BioVendor R&D Products, Czech Republic. The limits of individual adipokine detection were: lipocalin-2 0.02 ng/mL, chemerin 0.1 ng/mL, apelin 8.25 pg/mL, and adiponectin 26 ng/mL. The coefficients of variation within and between assays (CVs) were, respectively, for lipocalins-2 7.0% and 9.8%, chemerins 5.1% and 8.6%, apelins < 10% and <12%, adiponectin 4.9% and 6.7%.

The results obtained were necessary to demonstrate the relationship between the levels of the analysed adipokines, BMI at the first prenatal visit, and the BMI level at the time of sampling. In addition, attempts were made to correlate the concentrations of adipokines with the pregnant woman’s age, gestational age at the time of sampling, estimated foetal weight in ultrasound and OGTT blood glucose levels.

The following computer programs were used for statistical analysis: MedCalc version 15.8 PL and Statistica version 13 PL. The normality of the distribution of variables was tested using the D’Agostino–Pearson test. Due to the non-normal distribution of the assessed variables, non-parametric tests were used in further analysis. The non-parametric U-Mann–Whitney test (continuous variables) was applied to evaluate differences between clinical factors and concentrations of tested adipokines in the study and control groups. The Chi-square test assessed differences between clinical factors in data distribution that were not presented on an ordinal scale. The non-parametric Spearman’s rank correlation test was used to assess simple correlations between clinical factors and the concentrations of adipokines in the study group.

ROC curves were applied to assess the diagnostic usefulness of the tested adipokines in detecting gestational diabetes. The results with α probability values below 0.05 were considered statistically significant in all cases.

## 5. Conclusions

Gestational diabetes mellitus affects a large number of pregnant patients. The significance of adipokines in the pathophysiology of GDM is not fully clear, and none of the adipokines have been confirmed as an early biomarker for GDM screening. Identification of patients at risk of developing GDM already in the first trimester of pregnancy would make it possible to minimize maternal and neonatal morbidity and mortality. It also influences the later development of the offspring as well as the risk of future maternal morbidity.

The results of the presented study suggest that disturbances in adiponectin, lipocalin, chemerin and apelin production could be a part of the mechanisms involved in GDM pathogenesis. The GDM patients were found to have lower levels of adiponectin and higher levels of lipocalin, chemerin and apelin than non-GDM patients, which could result from impaired synthesis or release of these adipokines. It might be speculated that these adipokines could be considered as a new biomarker for predicting and early diagnosing GDM.

The main limitation of this study is that the number of patients was relatively small and homogenous. Therefore, the statistical analysis of the results could be underpowered. Further and more extensive prospective studies investigating the role of these adipokines in the pathogenesis and early prediction of GDM are required.

## Figures and Tables

**Figure 1 ijms-25-00175-f001:**
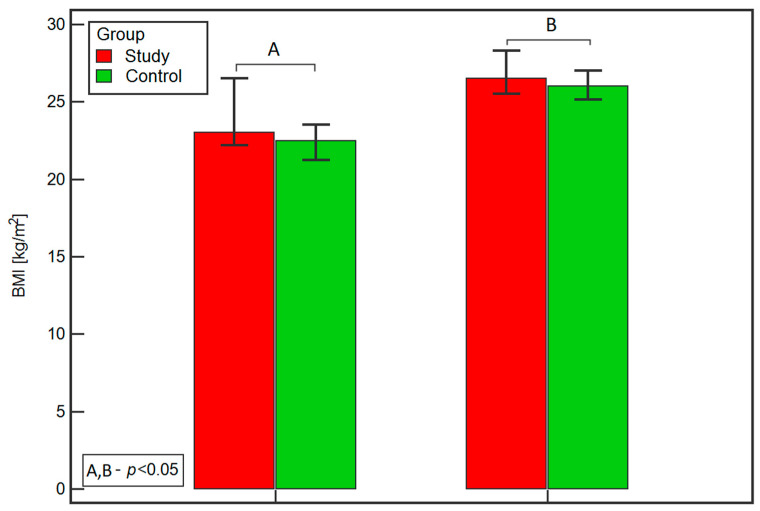
Comparison of BMI at the first prenatal visit (**A**) and at 24–28 weeks of pregnancy (**B**) between the study and control groups.

**Figure 2 ijms-25-00175-f002:**
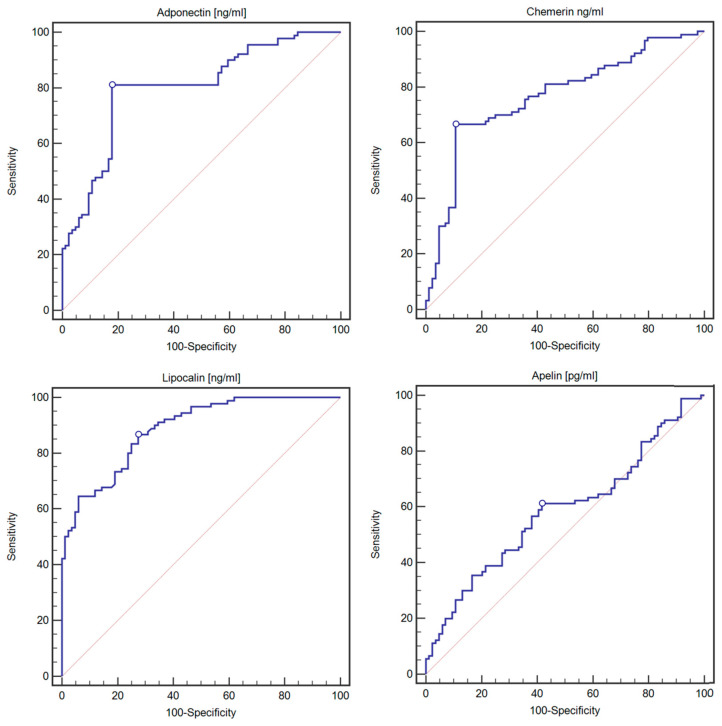
ROC curves representing the diagnostic usefulness of adiponectin, chemerin, lipocalin and apelin in detecting GDM.

**Table 1 ijms-25-00175-t001:** Clinical characteristics and comparison of the study group (GDM) and the control group.

Variable	Study Group—GDM (n = 90) Median (Interquartile Range)	Control Group (n = 84) Median (Interquartile Range)	*p*
Age	26.0 (23.0–31.0)	27.0 (24.0–31.0)	0.9278
BMI (first prenatal visit)	23.0 (22.2–26.5)	22.5 (21.2–23.5)	0.0356
BMI (24–28 weeks)	26.5 (25.5–28.3)	26.0 (25.1–27.0)	0.0337
Pregnancy			0.7201
1	36 (40%)	33 (39.3%)	
2	34 (37.8%)	30 (35.7%)	
3	14 (15.6%)	17 (20.2%)	
4	5 (5.6%)	2 (2.4%)	
5	1 (1.1%)	2 (2.4%)	
Gestational age (weeks)	26.5 (25.3–27.5)	26.6 (25.8–27.8)	0.3371
Foetal weight (g)	909.0 (735.0–1093.0)	956.0 (859.0–1106.0)	0.1429
TTG 0 (mg/dL)	94.0 (89.0–98.0)	81.0 (75.5–87.0)	<0.0001
TTG 1 (mg/dL)	185.0 (176.0–197.0)	138.0 (120.5–157.0)	<0.0001
TTG 2 (mg/dL)	155.0 (142.0–169.0)	127.5 (105.5–135.5)	<0.0001

**Table 2 ijms-25-00175-t002:** Comparison of the tested adipokines concentration in the study and control group.

Variable	Study Group—GDM (n = 90) Median (Interquartile Range)	Control Group (n = 84) Median (Interquartile Range)	*p*
Adiponectin [ng/mL]	7234.6 (5432.2–8393.2)	9837.5 (8694.7–14,752.9)	<0.0001
Chemerin [ng/mL]	264.0 (220.7–297.2)	206.7 (181.9–237.6)	<0.0001
Lipocalin [ng/mL]	39.5 (25.9–50.9)	19.4 (14.7–24.8)	<0.0001
Apelin [pg/mL]	11392.0 (7398.5–16,412.6)	9364.0 (7467.3–13,272.5)	0.0554

**Table 3 ijms-25-00175-t003:** Spearman’s rank correlations between adiponectin and selected variables in the study group.

Variable	n	Adiponectin [ng/mL]	
		rho	*p*
BMI (first prenatal visit)	90	−0.917	<0.0001
BMI (24–28 weeks)	90	−0.836	<0.0001
Pregnancy number	90	−0.297	0.0045
Gestational age	90	−0.092	0.3903
Foetal weight	90	−0.050	0.6402
TTG 0	90	−0.387	0.0002
TTG1	90	−0.616	<0.0001
TTG2	90	−0.655	<0.0001
Chemerin [ng/mL]	90	−0.805	<0.0001
Lipocalin [ng/mL]	90	−0.472	<0.0001
Apelin [pg/mL]	90	0.073	0.4958

rho—Spearman’s rank correlation measure, TTG0—fasting glucose, TTG1—glycemia 1 hour after OGTT, TTG2—glycemia 2 h after OGTT.

**Table 4 ijms-25-00175-t004:** Spearman’s rank correlations between chemerin and selected variables in the study group.

Variable	n	Chemerin [ng/mL]	
		rho	*p*
BMI (first prenatal visit)	90	0.749	<0.0001
BMI (24–28 weeks)	90	0.662	<0.0001
Pregnancy number	90	0.182	0.0856
Gestational age	90	0.010	0.9227
Foetal weight	90	−0.008	0.9378
TTG 0	90	0.461	<0.0001
TTG1	90	0.762	<0.0001
TTG2	90	0.746	<0.0001
Chemerin [ng/mL]	90	N/a	N/a
Lipocalin [ng/mL]	90	0.621	<0.0001
Apelin [pg/mL]	90	0.073	0.4958

rho—Spearman’s rank correlation measure, TTG0—fasting glucose, TTG1—glycemia one hour after OGTT, TTG2—glycemia 2 h after OGTT.

**Table 5 ijms-25-00175-t005:** Spearman’s rank correlations between LCN2 and selected variables in the study group.

Variable	n	Lipocalin [ng/mL]	
		rho	*p*
BMI (first prenatal visit)	90	0.390	0.0001
BMI (24–28 weeks)	90	0.304	0.0035
Pregnancy number	90	0.184	0.0826
Gestational age	90	0.018	0.8685
Foetal weight	90	0.053	0.6175
TTG 0	90	0.212	0.0449
TTG1	90	0.428	<0.0001
TTG2	90	0.565	<0.0001
Chemerin [ng/mL]	90	N/a	N/a
Lipocalin [ng/mL]	90	N/a	N/a
Apelin [pg/mL]	90	−0.058	0.5888

rho—Spearman’s rank correlation measure, TTG0—fasting glucose, TTG1—glycemia one hour after OGTT, TTG2—glycemia 2 h after OGTT.

**Table 6 ijms-25-00175-t006:** Spearman’s rank correlations between apelin and selected variables in the study group.

Variable	n	Apelin [pg/mL]	
		rho	*p*
BMI (first prenatal visit)	90	−0.166	0.1169
BMI (24–28 weeks)	90	−0.193	0.0684
Pregnancy number	90	−0.006	0.9578
Gestational age	90	−0.100	0.3485
Foetal weight	90	−0.118	0.2697
TTG 0	90	0.084	0.4329
TTG1	90	0.025	0.8138
TTG2	90	−0.095	0.3709
Chemerin [ng/mL]	90	N/a	N/a
Lipocalin [ng/mL]	90	N/a	N/a
Apelin [pg/mL]	90	N/a	N/a

rho—Spearman’s rank correlation measure, TTG0—fasting glucose, TTG1—glycemia one hour after OGTT, TTG2—glycemia 2 h after OGTT.

**Table 7 ijms-25-00175-t007:** Evaluation of the diagnostic usefulness of the tested adipokines in the detection of gestational diabetes using ROC curves.

Adipokine	AUC [95% CI]	Sensitivity (%)	Specificity (%)	Cut-Off Point	*p*
Adiponectin [ng/mL]	0.801 [0.733–0.857]	81.11	82.14	≤8482.06	<0.0001
Chemerin [ng/mL]	0.767 [0.697–0.827]	66.67	89.29	>249.99	<0.0001
Lipocalin [ng/mL]	0.887 [0.830–0.930]	86.67	71.62	>23.69	<0.0001
Apelin [pg/mL]	0.584 [0.507–0.658]	61.11	58.33	>9888.2	0.0529

AUC—area under curve, CI—confidence interval.

## Data Availability

The data used to support the findings of this study are included within the article.

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
