# Peer review of "Comparative Evaluation of Adipokine Metrics for the Diagnosis of Gestational Diabetes Mellitus"

_ijms, 2023, doi:10.3390/ijms25010175_

Round 1

Reviewer 1 Report

Comments and Suggestions for Authors

I would suggest that the introduction be shortened so as to emphasize what is known so far and what is not known enough on adipokines in GDM.

In the results section, the authors should not repeat in the figures what is already shown in the tables.

The discussion section should start with the main findings of the study. The authors should avoid repetition of the obtained results (numbers and p-values) and what was already said in the introduction.

The authors should consider changing the title of the manuscript to indicate that this is an experimental paper.

Author Response

We would like to thank the Reviewer for careful and thorough reading of this manuscript.

According to Reviewer’s suggestions:

  1.     We tried to shorten the “Introduction”. However, due to the profile of the journal, we decided to leave the discussion of the molecular mechanisms of action of adipokines and according to the second Reviewer suggestions we added some sentences.
  2. We omitted the Figure 2 because the same data are presented in the Table.
  3. In the Discussion, in each of the paragraphs discussing the results regarding individual adipokines, we start by presenting the results of our research. We also removed the obtained results (numbers and p-values) and what was already said in the introduction.
  4. We changed the title of our article (this title was suggested by Second Reviewer).

Once more we would like to thank the Reviewer for the excellent work.

Reviewer 2 Report

Comments and Suggestions for Authors

The review article "Possible role of adipokines in the pathogenesis and diagnosis of gestational diabetes mellitus" results showed significant differences in the levels of adiponectin, chemerin, and lipocalin between pregnant women with gestational diabetes mellitus (GDM) and healthy ones. 

Overall, the paper has a solid structure and organized relevant biomedical content. The writing is thoughtful and in-depth, offering a detailed description of each scientific subject discussed in your assigned research article manuscript.

Here are some issues that might need attention:

1. Title: The title could be more concise and give a clear image of the novelty that the research brings into the biological field. Something like Comparative evaluation of adipokine metrics for the diagnosis of gestational diabetes mellitus" could be better.

2. To date, there are more than 50 different adipokines known to function in various aspects of diabetes and associated complications. What are the criteria for selecting these four different adipokines only? The authors must add a paragraph on the contrary function of adiponectin and chemerin (the author must cite the articles that are very closely related to the MS: PMID: 29669464 and 38004353) during metabolic conditions including gestational diabetes. 

3. The authors have started with insulin resistance in the introduction and its correlation with GD. However, there is no data on fasting insulin levels or HOMA-IR to show in the article. Did the authors check that parameter and try to correlate it with the results "Adiponectin/chemerin"

4. Contents Structure and Organization: Each paragraph is set orderly with lists for fine organization. However, some paragraphs exceed the typical length.

  1. Provide comprehensive overview for every studied variable's interaction in path renewing related complications.
  2. Could also emphasize more on comparisons between complex functions of these variables in differing stages of GDM.

1.      Errors in content: I didn't find any factual errors since all statements supported by well-based results, conclusions and appropriately cited.

2.      Figures and Tables: Appropriately used and well-labelled.

3.      Writing style: Somewhat complex and difficult in certain instances (lines 773-784, 1104-1108, 849-856). It could be helpful to simplify the language or sentence structure.

Author Response

We would like to thank the Reviewer for careful and thorough reading of this manuscript.

According to the Reviewer’s suggestions:

  1. We changed the title of our article (it was also suggested by First Reviewer).
  2. We added a paragraph explaining the criteria for selecting these four adipokines. Indeed, as the Reviewer noted, there are more than 50 different adipokines known to function in various aspects of diabetes and associated complications. However, not all appear to play the significant role in the pathogenesis of GDM. We also introduced the paragraph regarding the contrary function of adiponectin and chemerin and we cited suggested by the Reviewer valuable references.
  1. Because we were not able to obtain the data on the insulin levels of all patients, we did not present this analysis in our study. However, in the study group and in the control group there were no obese patient with BMI > 30 kg/m2. It is important, that the highest BMI value at the first prenatal visit was 26.5 kg/m2 in GDM group and 23.5 kg/m2in control group. We excluded from our study the patient with pre-pregnancy diabetes mellitus, insulin resistance diagnosed before pregnancy, metabolic disorders (such as polycystic ovary syndrome), and any form of hypertension. So, the risk of markedly higher insulin resistance in our study group in comparison with control group was relatively small. However, the Reviewer’s comment on the need of the including this parameter is very valuable and we will gladly observe it in our further studies.

4.     We removed the phrase “related complications” from the Abstract because we actually focused on the pathogenesis and screening of GDM and it was misleading. In the Discussion of most of the works, we emphasized the period of pregnancy and GDM during which the studies were performed.

5.     The last point is a bit unclear to us. There are only 885 lines in the article, of which from 641 are references. Therefore, we do not know how to respond to possible corrections in the lines suggested by the Reviewer. 

Once more we would like to thank the Reviewer for the excellent work.

Round 2

Reviewer 2 Report

Comments and Suggestions for Authors

Thank you authors for justifying the comments raised by me and incorporating the changes at specific positions.